# Shell dissolution observed in *Limacina helicina antarctica* from the Ross Sea, Antarctica: paired shell characteristics and in situ seawater chemistry

Kevin M. Johnson<sup>1</sup>, Umihiko Hoshijima<sup>1</sup>, Cailan S. Sugano<sup>1</sup>, Alice T. Nguyen<sup>1</sup> and Gretchen E. Hofmann<sup>1</sup>

Department of Ecology, Evolution and Marine Biology, University of California, Santa Barbara, Santa Barbara CA 93106-9620 USA

*Correspondence to*: Gretchen E. Hofmann (gretchen.hofmann@lifesci.ucsb.edu)

**Abstract**. The euthecosome (shelled) Antarctic pteropod, *Limacina helicina antarctica*, is a dominant member of the Southern Ocean macrozooplankton community, and due to its aragonitic shell, is thought to be particularly vulnerable to ocean acidification and undersaturation conditions that are predicted in the future. Notably, pteropods in surface waters and

- near the continental shelf in the Ross Sea are highly vulnerable as these regions are predicted to be seasonally under-saturated within 2-3 decades. Carbonate chemistry data are rare for this region and here we present the results of a 6-week field study and report patterns of dissolution of juvenile pteropods along with carbonate chemistry of seawater at the time of collection. Conducted in McMurdo Sound in the south Ross Sea in the Pacific sector of the Southern
- Ocean, *L. h. antarctica* was successfully collected in plankton tows through the fast sea ice at a single station at 50 m. During the 6-week field study, ocean pH was relatively stable, ranging from 7.988 in October to 8.029 by early December. Calculated saturation states for aragonite ( $\Omega_{arag}$ ) over the 6-week study period ranged from 1.16 to 1.24. Pteropods collected at each sampling time point were prepared for SEM and analysis revealed that roughly 63% of the shells
- displayed some degree of shell irregularities suggesting that active dissolution of the aragonitic shell was ongoing under in situ conditions. These results add to the accumulating evidence that shelled pteropods of the Southern Ocean are experiencing aragonite under-saturation events in the present-day that lead to a majority of individuals displaying shell dissolution. Predicted changes to the carbonate system in the Southern Ocean from ocean acidification will likely

expand the intensity and duration of these under-saturation events, increasing the need to better understand how pteropods will fare in response to ocean acidification.

**Keywords**: *Limacina helicina antarctica*, pteropod, ocean acidification, Ross Sea, pH, Southern Ocean

5 Ocean

# **1 Introduction**

- The uptake of anthropogenic CO<sub>2</sub> by the surface ocean, a process known as ocean
  acidification, is reducing the pH and carbonate mineral saturation state of seawater at an unprecedented rate ((Feely et al., 2004;Fabry et al., 2009;Orr et al., 2005). The euthecosome (shelled) pteropods, such as *Limacina helicina*, precipitate a comparatively soluble form of calcium carbonate (aragonite), leading to increased sensitivity to changes in carbonate chemistry (Bednaršek et al., 2012a;Fabry, 1989). Recently, pteropods in the *Limacina* complex
- have been identified as organisms that are highly vulnerable to ocean acidification and thus may serve as an indicator organism for trajectories of change in high latitude marine ecosystems (Fabry et al., 2009;Bednaršek et al., 2016a;Comeau et al., 2009;Lischka et al., 2011). In this study, we report on the results of a field study in McMurdo Sound in the Pacific sector of the Southern Ocean where shelled Antarctic pteropods, *Limacina helicina antarctica*, were collected
- during the 2014 austral summer. In presenting paired organismal data and in situ measurements of pCO<sub>2</sub> at the time of collection, this study illustrates how present-day carbonate chemistry may present a challenge to biogenic calcification in the Antarctic pteropod, *L. h. antarctica*.

The Southern Ocean is predicted to be first in time to experience changes in carbonate chemistry due to ocean acidification (McNeil and Matear, 2008;Fabry et al., 2009;Orr et al., 2005;Hauri et al., 2015). Recent modeling efforts in the Southern Ocean have underscored the extent to which aragonite undersaturation events are occurring and will occur due to ocean acidification. Hauri et al. (2015) used a suite of Earth system models to forecast that by the year 2100, surface waters in the Southern Ocean will be undersaturated with respect to aragonite for

six months out of the year. In McMurdo Sound, Kapsenberg et al. (2015) conducted a detailed

investigation into the near-shore Antarctic carbonate chemistry system using autonomous pH sensors and proposed that McMurdo Sound may experience under-saturation during the austral winter by the year 2030. With increasing anthropogenic CO<sub>2</sub> levels, the saturation state of calcium carbonate minerals in many marine ecosystems is decreasing, thereby challenging

calcification in a number of species (Hofmann et al., 2010). With shells composed entirely of aragonite, pteropods in the *Limacina* complex may act as sentinel organisms for the impacts of ocean acidification (Bednaršek et al., 2014a;Bednaršek et al., 2016b;Peck et al., 2015).

Although pteropods are held up as bioindicators of future change in ocean chemistry, recent assessments using scanning electron microscopy (SEM) of *Limacina spp*. in the Eastern

- Pacific and Southern Ocean suggest that shell dissolution, driven by seasonal patterns of upwelling in coastal regions, are already occurring (Bednaršek et al., 2012; Bednaršek et al., 2014). With regard to *L. h. antarctica*, investigators have detected shell dissolution in recently collected field samples (Bednaršek et al., 2012). Although there is some suggestion that these dissolution patterns only occur when the protective outer organic layer (i.e., the periostracum)
- has been breached by some physical process (Peck et al., 2015). Nevertheless, pteropod shells are composed of aragonite, a comparatively soluble form of calcium carbonate (Andersson et al., 2008), and likely susceptible to dissolution in nature when exposed to undersaturated seawater (Bednaršek et al., 2014b;Fabry et al., 2008;Fabry et al., 2009). Although researchers have made shipboard collections and analyzed preserved shells of *Limacina* from the Antarctic,
- few studies have examined the integrity of the shell when specimen collection is paired with in situ seawater chemistry. Further, given the scarcity of data on the seawater chemistry of nearshore polar waters (Schram, 2015;Kapsenberg et al., 2015), there is a discrepancy in our understanding of the current conditions that *L. h. antarctica* encounters in the water column.

In this light, the goal of this study was threefold: (i.) to collect pteropods in the field and determine the pH and saturation state of the seawater at the time of collection in the austral summer, (ii.) to describe basic organismal and ecological traits such as size and abundance of the collections, and (iii.) use SEM to assess patterns of shell conditions in collected specimens. These analyses are the first extended pteropod sampling effort aimed at capturing the combined natural variability of seawater chemistry and shell conditions in Antarctic pteropods. The

results from this study add to a growing body of literature that highlights pteropods as sentinel organisms of ocean acidification.

# 2 Materials and methods

5

# 2.1 Field sampling of pteropods

Juvenile pteropods (*Limacina helicina antarctica*) were collected through the sea ice in McMurdo Sound in the southern Ross Sea during the austral spring and summer, from OctoberDecember 2014. Pteropod collections were performed at a depth of 50 m via plankton tows

- performed by hand at a rate of approximately 2.4 m s<sup>-1</sup>, with a small collapsible plankton net (100  $\mu$ m mesh) designed specifically to be deployed through small-diameter holes drilled through the sea ice (Sewell, 2005). Plankton tows were performed at least twice per week at a marked location near Cape Royds (77°33.9' S, 166°11.23333' E) (see Fig. 1). Each sampling
- 15 event consisted of three consecutive tows to 50 m. For each tow, the number of *L. h. antarctica* were individually enumerated and field densities were then calculated by dividing the total number of pteropods by the total volume of water sampled (Hunt et al., 2008). Water chemistry samples were taken following the recovery of every tow at a depth of 50 m using a Niskin sampler, transported back to McMurdo station, and immediately preserved with saturated
- 20 mercuric chloride according to Dickson et al. (2007).

# 2.2 Post-collection handling

After collection, pteropods were maintained in ambient seawater (-1.9 °C) in 500 ml 25 Nalgene containers, and immediately transported to the McMurdo laboratory facilities. Animals from each collection were counted in order to estimate catch effort. To prepare samples for morphological measurements and scanning electron microscopy (SEM), pools of 10 pteropods were incubated in 50% ethanol overnight at + 4 °C and then transferred to 70 % ethanol and stored at + 4 °C.

#### 2.3 Methodology for scanning electron microscopy

Scanning electron micrographs were obtained for closer examination of shell surface conditions at five collection time points that spanned the 6-week study period. To remove abiogenic

- crystals from the shell surface, pteropod samples were gently rinsed with fresh 70 % ethanol twice followed by a brief rinse in 95 % ethanol. Samples were subsequently air-dried before using fine brushes to carefully position shells in the apical view on aluminum stubs mounted with adhesive carbon conductive tape. To avoid causing structural damage to the apical side of the shell (i.e., the side that was imaged), specimens were positioned using two fine brushes that
- only came into contact with the basal side of the shell. Mounted specimens were then sputtercoated with gold/palladium for 90 seconds and imaged with a FEI Inspect S Electron Scanning Microscope at the Microscopy and Microanalysis Facility of the Materials Research Laboratory (MRL) Shared Experimental Facilities located at UC Santa Barbara.

#### 15 2.4 Analysis of shell dissolution

Semi-quantitative analysis of shell dissolution was performed using the same language and patterns of dissolution as described in Bednaršek et al. (2012b). We categorized three types of dissolution that are reported as minor (Type 1), moderate (Type 2), and severe (Type 3) (see

- Fig. 6). The analysis also scored shells without signs of shell irregularities as a Type 0 pattern. To standardize the total area of the shell analyzed between individuals, percent cover of each dissolution type was measured over the protoconch region of each shell using SEM imagery and the program ImageJ (1.49v). Percent cover data of each dissolution type was compared within tows using a Kurksal Wallis test and a Dunn test of multiple comparisons was used to test for
- differences between tows (Bednaršek et al., 2012c). The number of shells analyzed and corresponding date of field collection is as follows: tow 2 on November 4 (n = 11); tow 7 on November 18 (n = 11); tow 8 on November 22 (n = 7); tow 9 on November 24 (n = 10); and tow 14 on December 8 (n = 12).

#### 30 2.5 Morphological measurements

For each tow, 30 individuals were imaged using a Cannon Powershot A630 mounted on a Wild M37Z stereomicroscope for morphological measurements. To estimate size, shell diameter was measured along a line 90° from the apex of the protoconch to the edges of the shell on a flat-lying specimen (Fig. 2).

#### 2.6 Carbonate chemistry

Seawater chemistry was performed on samples collected form the field at the McMurdo Station Crary Laboratory. Bottle samples, collected in borosilicate bottles, were preserved with saturated mercuric chloride immediately upon arrival at McMurdo Station, and subsequently

- stored at + 4 °C until analyzed as described in Standard Operating Procedure 1 (SOP1) (Dickson et al., 2007). From each sample, pH was measured using the m-cresol (Sigma-Aldrich®) spectrophotometric method described in SOP 6b (Dickson et al., 2007). Salinity was measured using a calibrated YSI 3100 conductivity probe and total alkalinity measurements were conducted using open-cell titration method (SOP3b) with a Mettler-Toledo T50 (Dickson
- et al., 2007). To calculate field pH and aragonite saturation states, the computer software CO2Calc was used with the in situ temperature estimated to be -1.9 °C based on multi-year pH and temperature time-series data (Kapsenberg et al., 2015;Robbins et al., 2010).

#### **3 Results**

#### 20 3.1 Pteropod collections

Pteropods were collected at a study site near Cape Royds for approximately six weeks, from October 31<sup>th</sup> -December 8<sup>th</sup> 2014 (Fig. 1). Plankton towing at this location began on October 29th, with the first collection of pteropods occurring on October 31. *L. h. antarctica* individuals were consistently collected from that date until the last tow on December 8th, 2014

(a date that was set by the closing of the sea ice near McMurdo Station).

Over the collection period, *L. h. antarctica* densities ranged from 3 - 81 individuals m<sup>-3</sup> and reached peak abundances between November 11<sup>th</sup> and November 24<sup>th</sup> (Fig. 3). Body size of *L. h. antarctica* ranged from 0.484 mm - 1.54 mm and were distributed amongst three distinct size classes: 0.84  $\pm$  0.09 mm, 1.01  $\pm$  0.15 mm, and 1.19  $\pm$  0.14 mm; Tukey's Honestly

Significant Difference (HSD) test (adjusted p-value

#### 3.2 Paired in situ carbonate chemistry

Over the course of the collection period, carbonate chemistry remained relatively stable with slight changes towards the end of the 6-week period (Fig. 5). Average pH for Cape Royds seawater samples on October  $31^{st}$  was 7.988 ± 0.003, and increased to 8.029 ± 0.003 by

5 December 5<sup>th</sup>, 2014. The aragonite saturation state of the seawater ( $\Omega_{arag}$ ) on October 31<sup>st</sup> was 1.159 ± 0.0067 and increased to 1.239 ± 0.002 by December 5<sup>th</sup>, 2014. Overall, by December, Cape Royds had experienced a decrease in pCO<sub>2</sub> of 41 µatm and a modest increase in  $\Omega_{arag}$  of 0.08. A complete list of carbonate chemistry for each sampling event is shown in Table 1.

#### 10 3.3 Shell characteristics and dissolution patterns

Dissolution was observed in a majority of pteropods shells from all collection periods. The spatial extent of this damage, classified by varying degrees of dissolution (illustrated in Fig. 6), varied both within and between collections.

15

- 5 Overall, of the 51 individual pteropod shells analyzed over the five collections, 63 % of the shells exhibited some form of dissolution on the protoconch. There was a pattern of linkage amongst the shell dissolution types. Specifically, among the 51 samples, 14 % (n = 7) of shells showed severe dissolution, and all of these also showed moderate dissolution; 29 % (n = 15) of shells showed moderate dissolution as the worst shell condition and all but one of these also
- 20 showed minor dissolution; 20 % (n = 10) of shells exhibited minor dissolution as the worst shell condition. Finally, there were many samples that displayed no signs of shell irregularities. Here, 37 % (n = 19) showed no signs of minor, moderate, or severe dissolution.

In terms of gross shell morphometrics, pteropod shells in our collections varied in total protoconch area (7,401 ~ 13,906  $\mu$ m<sup>2</sup>, mean ± s.d. = 9,387 ± 2,532  $\mu$ m<sup>2</sup>), but this did not vary

25 significantly between collection days (1-way ANOVA, p = 0.55). The extent of total dissolution coverage on the protoconch was widely variable as well. While 80 % (n = 41) of pteropods exhibited dissolution on less than 50% of their total protoconch, 0 % of pteropods exhibited moderate and severe dissolution over more than 50 % of their total area.

Shell dissolution classes exhibited variation both within and between collections (Fig. 7).30 Significant differences were observed in protoconch coverage of undamaged and minor

dissolution between tows (Kurksal – Wallis test, p = 0.046 and p = 0.012, respectively). These differences are largely attributed to Tow 14, which showed a relatively low undamaged area and relatively high Type 1 area (Dunn test of multiple comparisons, see Fig. 7).

To further compare shell conditions between tows, shells were also classified by the 5 worst dissolution class observed (Fig. 8). Overall, tows were not significantly different when analyzed in this manner (Fisher's Exact Test, p = 0.139). Thus, the variation between tows resides in the ratio of minor dissolution classes rather than in the most severe class of dissolution observed.

#### 10 4 Conclusions

Recently, pteropods have been held up as a potential indicator organism for the biological impacts of ocean acidification because the aragonitic shell is comparatively vulnerable to dissolution. Previous studies that collected animals on cruises and reported a high

- prevalence of shell dissolution set the stage to link shell abnormalities to exposure to undersaturated seawater with respect to aragonite. The results of this study corroborate this body of earlier work, and shows that some fraction of the collected animals displayed patterns of porosity and corrosion indicative of shell dissolution, although we also collected large numbers of animals with apparently undamaged shells. In addition, the seawater chemistry
- performed on bottle samples taken simultaneously with the plankton tows indicate that *Limacina* experienced seawater conditions where saturation state was quite low, although not technically undersaturated ( $\Omega_{arag} = 1.159 \pm 0.0067$ ). Overall, our results indicate that a high percentage of this macrozooplankton is experiencing shell loss in situ under present-day carbonate chemistry conditions in the Ross Sea.
- Laboratory experiments conducted on this same research project showed that holding Limacina in high pCO<sub>2</sub> (900 μatm) for 14 days resulted in significant shell dissolution. Although this is not a calibration of dissolution patterns to in situ conditions, it is direct evidence that exposure of live Antarctic pteropods to undersaturated seawater conditions was sufficient to cause shell irregularities (K.M. Johnson, unpublished observation). Previous in situ studies of
- shell dissolution in Antarctic pteropods found high levels of dissolution across six locations in

the Scotia Sea (Bednaršek et al., 2012). Additional perturbation experiments using pteropods from the Scotia Sea also documented net dissolution when *L. h. antarctica* was reared under high pCO<sub>2</sub> conditions (Bednaršek et al., 2014b). Similarly, research on Arctic populations of *L. helicina* showed reduced growth rate, dissolution and increased mortality under high pCO<sub>2</sub>

treatments after 28-days (Lishcka et al., 2010).

While the actual exposure conditions or natural events that cause dissolution are still sometimes debated (Bednaršek et al., 2016b;Peck et al., 2015), it is clear from the collections we have made in McMurdo Sound that *L. h. antarctica* experiences conditions in the water column that drive shell dissolution. As a result of the sensitivity of the pteropod shell to

- carbonate chemistry conditions, pteropod species have been suggested to be sentinel organisms (Roberts et al., 2011;Bednaršek et al., 2016a;Bednaršek and Ohman, 2015). The shell itself is indeed an ideal structure to monitor as it is a thin, aragonitc shell, with multiple layers (Sato-Okoshi et al., 2010). Nano-indentation techniques have been developed to test the physical strength and hardness of the *L. h. antarctica* shell (Teniswood et al., 2013), with the intention of
- linking shell strength to exposure to undersaturated waters in situ. Further, these nanomaterials techniques were used to compare the shell properties of *L. h. antarctica* collections that are almost a decade apart (Teniswood et al., 2016) with some differences being observed. Challenges in the use of *Limacina* shells as sentinel, bio-recording structures remain because it is not know what carbonate chemistry the pteropods were exposed to in situ prior to collection.
- Indeed in the present study, more than half the shells we examined displayed shell irregularities, but it is unclear as to what depth or season caused the observed dissolution of juvenile shells.

*L. h. antarctica* has been the subject of research that examined physiological responses to experimentally altered carbonate chemistry conditions. Previous studies have found that  $CO_2$  exposure suppressed metabolism in years when phytoplankton abundance is high, but does not

significantly impact metabolism in years of low phytoplankton biomass (Seibel et al., 2012). L. h. antarctica metabolism was also measured in response to low food availability (Maas et al., 2011). Our own studies of juvenile pteropods found that L. h. antarctica exhibited a differential response to pCO<sub>2</sub> only under ambient temperatures (-1.0 °C) (Hoshijima et al., 2016). Specifically, pteropods exposed to the high pCO<sub>2</sub> treatment exhibited elevated respiration rates

relative to those exposed to low  $pCO_2$  conditions. However, this  $pCO_2$  treatment effect was not evident when pteropods experienced elevated  $pCO_2$  at an elevated temperature of +4 °C.

Recent research focused on the impacts of ocean acidification on other Antarctic marine invertebrates, in general, has shown a range of responses, with most of the described responses

- being deleterious. For other members of the zooplankton, laboratory studies have shown that early stage krill (*Euphausia superba*) are sensitive to high pCO<sub>2</sub> levels (Kawaguchi et al., 2011). Risk maps based upon the sensitivity of krill egg hatching success have also been developed for regions of the Southern Ocean (Kawaguchi et al., 2013) and suggest that high-risk areas for krill will develop in regions such as the Weddell Sea. Similarly, studies on adult forms of *E. superba*
- have measured metabolic changes that indicate elevated pCO<sub>2</sub> induced a cost to maintenance (Saba et al., 2012). Data on Antarctic benthic marine invertebrates show similar responses. For example, a 6-week study on the effects of ocean acidification and temperature on adult Antarctic limpets, *Nacella concinna*, and snails, *Margarella antarctica*, found negligible impacts on *M. antarctica* and a small decrease in gonadal lipid content in *N. concinna* (Schram et al., 2012).
- 2016). Taken together these results suggest Antarctic benthic invertebrates may have a significant level of resistance to changes in pH.

Overall, these responses are critical in light of recent projections regarding Southern Ocean carbonate chemistry that suggest that by 2050, surface waters will be undersaturated with respect to aragonite for multiple months each year and will reach close to yearlong

undersaturation by 2100 (Hauri et al., 2015) and seasonal undersaturation will be an issue even sooner (Kapsenberg et al., 2015). These projections, coupled with research into the contemporary carbonate system in the Southern Ocean have highlighted the extent to which undersaturation events are occurring and will occur in the future.

The observed levels of dissolution found in this study adds to the growing body of

- evidence that find contemporary shell dissolution in the globally distributed pteropod, *L. helicina*. This pattern raises important questions as to the effect dissolution has on organismal performance, fitness and future viability. In addition, the risk to populations of pteropods will be defined by the impacts of OA on reproductive success. Although we have little information about how variable carbonate chemistry influences early development in situ, laboratory studies
- have shown that exposure to high pCO<sub>2</sub> levels can lead to decreases in egg densities in gravid

females with up to 80% of those eggs failing to successfully develop into larvae (Manno et al., 2016). These results highlight the need for further research into how *L. h. antarctica* and other key macrozooplankton species will fare in response to multiple stressors in the Southern Ocean.

# 5 Data availability

Following publication, Dryad (<u>http://datadryad.org/</u>) will be used as a repository for the organismal and carbonate chemistry data in this study.

#### Author contributions

10 KMJ, OH, and GEH conducted the fieldwork (plankton tows and water sampling), and processed samples at McMurdo Station. OH and KMJ performed carbonate chemistry measurements and calculations. KMJ, ATN and CSS performed the scanning electron microscopy; CSS and OH performed the analysis of the SEM data. Finally, all authors contributed to writing the manuscript.

#### 15

# **Competing interests**

The authors declare that they have no conflict of interest.

# Acknowledgements

- The authors would like to thank members of the U.S. Antarctic Program and Lockheed Martin's Antarctic Support Corporation (ASC) for support at McMurdo Station, Antarctica during the 2014-2015 Summer Field Season. In addition, we thank Professor Mary Sewell (University of Auckland) for the loan of the collapsible plankton net, and Mr. Carl Reim (Polar Geospatial Center, University of Minnesota) for preparing the map of our study sites. This research was
- supported by a grant from the U.S. National Science Foundation (NSF) through the U.S. Antarctic Program (PLR-1246202 to GEH). During the course of this project, Kevin Johnson and Umihiko Hoshijima were each supported by an NSF Graduate Research Fellowship. The MRL Shared Experimental Facilities are supported by the MRSEC Program of the NSF under Award No. DMR 1121053; a member of the NSF-funded Materials Research Facilities Network
- (www.mrfn.org).

5

25

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

  - Kapsenberg, L., Kelley, A