# Peer review of "Shell dissolution observed in *Limacina helicina antarctica* from the Ross Sea, Antarctica: paired shell characteristics and in situ seawater chemistry"

_Biogeosciences, 2016_

## Referee Comment (RC1) · Anonymous Referee #1 · 21 Dec 2016

**Anonymous Referee #1**

Review for Biogeosciences, manuscript BG-2016-467 "Shell dissolution observed in Limacina helicina antarctica from the Ross Sea, Antarctica: paired shell characteristics and in situ seawater chemistry" by Johnson KM, Hoshijima U, Sugano CS, Nguyen AT, Hofmann GE

General comments

This manuscript aims at describing in situ carbonate chemistry conditions of the thecosome pteropod Limacina helicina antarctica in the Antarctic Ross Sea paired with their shell condition and provide some base data on species life history traits and abun-

dance. The motivation for this study comes on the one hand from the growing evidence of pteropods as sentinel organisms of ocean acidification, the fact that the study region (McMurod Sound) is proposed to experience aragonite undersaturation by the year 2030 during winter, and on the other hand from the lack of ecological knowledge on pteropods in the Ross Sea. Calculated aragonite saturation states ranged from 1.16 to 1.24, and hence were never below 1. Roughly 63% of investigated shells showed some degree of degradation (of the protoconch area). A main conclusion put forward in the abstract is that "shelled pteropods of the Southern Ocean are experiencing aragonite under-saturation events in the present-day that lead to a majority of individuals with shell dissolution".

Generally, I appreciate this type of approach looking at the in situ state of pteropods related to prevailing biogeochemistry and acquire more ecological data important to describe life history traits very much. All of this hasn't been done before in the Ross Sea (to my knowledge) and we are lacking knowledge on thecosomes in light of ongoing ocean change. However, the present manuscript has some major shortcomings and I don't think it can hold what it seems to promise at the beginning. Most importantly, I think the conclusion in the abstract mentioned above is not supported by the data because the authors don't know anything about the recent carbonate history of the pteropods but only assume that pteropods lived at under-saturated conditions and this being the reason for the dissolution patterns found. Further shortcomings are: 1) The shell dissolution was analyzed only in the protochonch area of the shells. The protoconch is the embryonal/veliger stage and oldest part of the shell. In many species it is naturally shed or broken off from the permanent adult shell (Lalli and Gilmer 1989). I.e. it is the area where damage due to any kind of reason first becomes apparent. No characterization of dissolution of the rest of the shell was done which makes it rather impossible to compare these data with other published work. In fact, the younger part of the shells (juvenile) were apparently not(?) affected by degradation (Fig. 6) indicating pristine shells, thus rather suggesting life at super-saturated $\Omega$aragonite conditions. 2) No attempt was made to relate biogeochemistry data to dissolutions patterns found,

i.e. shell dissolution results were analyzed isolated from prevailing pH, Ωaragonite or pCO2. That way it is not possible to draw any sound conclusion whether the described dissolution patterns have anything to do with carbonate chemistry and even more so with potential changes due to anthropogenic impact. However, as presently prepared, the manuscript "pretends" that results proved a connection between shell dissolution and in situ seawater chemistry. I don't think it allows for such conclusion. 3) Abundance and shell size data were collected but apart from the fact that they were collected and presented in the results, the reader is left alone with these data. Certainly it is not possible to deduce a whole life cycle from such data collected over a rather short time interval (6 weeks), but still it is possible to draw some conclusions and compare with current knowledge. What do these (precious indeed!) data tell us?

For these reasons, unfortunately, I cannot recommend the present manuscript for publication. I would really like to see a resubmission including an amended data set (with more elaborate shell analyses for example) and data analyses (linking shell dissolution to prevailing carbonate chemistry), if possible, and free of unfounded speculations.

Some minor comments:

- Introduction: Actually, the Arctic Ocean is expected to be first in temporal/spatial undersaturation events, not the Southern Ocean.

- M&M: Nutrient data are lacking in the carbonate chemistry calculations. Should be included as they affect results.

- Hoshijima et al. 2016: Is this an accepted manuscript in J Exp Biol? Couldn't find it?

---

## Referee Comment (RC2) · Anonymous Referee #2 · 23 Jan 2017

There is a growing body of papers linking carbonate chemistry and shell dissolution, firmly establishing the link between the carbonate chemistry and shell dissolution. While this manuscript only reestablishes this correlation, it does not contribute much to the development of new knowledge or approaches. Contrary of what the authors are claiming this being the first study to combine natural variability and shell dissolution, that is not the case, not it is presented as such. There are several issues with this manuscript that need to be consistently addressed. Firstly, the major drawback of the paper is the lack of the working hypotheses. That leaves the work more a compilation of different facts, without any proper integration. Authors have to reconsider how to bring

the results in the framework that addresses NEW questions and form the hypothesis around.

Secondly, this is a poorly designed and conceived study, with inconsistent methodological and statistical approaches that cannot be verified. The authors have obviously used a bit different approach in preparing the samples that is different from previous approaches. What was the reason for doing this? Was a method systematically tested and how was this done for the authors to be confident in the interpretation of their results? Did the periostracum stay intact or could this method have caused breaching of the periostracum at any stage or are the authors confident that no additional damaged were produced during the process? I would urge the authors to prepare and present additional results on pteropods that did not undergo this treatment but rather standardized treatment.

The authors did not have any controls in this study – I request for all the controls to be consistently presented in the paper.

I am perplexed on why the temperature of the water was not measured along with pH. Using temperature from the multi-year series for calculating carbonate chemistry parameters can have major drawbacks and error propagations. Given that temperature can significantly impact aragonite saturation state, it is an imperative to include standard error and uncertainty range in all carbonate chemistry parameters throughout the whole season of measured pH.

In regards to dissolution estimates, I cannot trace the tows to the carbonate chemistry conditions. Make that link more explicit. I wonder how the authors explain excessive presence of Type III dissolution when the organisms have been exposure to supersaturated conditions throughout the whole study. Is it possible that the damages actually originated before the samples were collected – during the austral winter? If that was the case then the carbonate chemistry presented does not really matter.

Where are the figures from Figure 6 from and when where they collected? Indicate

how many individuals per each tow look like 6a-c, d-f, g-j. Is image g-j all from one organism?

Please, clarify what additional information is captured in Figure 8 that has not been captured in Figure 7. If no different, remove Figure 8.

The authors have decided to analyze only the first whorl – why is that? I disagree with the authors that analyzing such a small surface can be unbiased and representative. While this is the easiest approach it does not give the full insight into dissolution. I would like to see the whole shell surface analyzed instead and presented. Only in such way, the results of this study can be comparable to the other studies.

The authors need to provide more discussion about the periostracum. I do not see any evidence that periostracum would be mechanically or chemically damaged – does this then mean that the dissolution is a results of carbonate chemistry conditions, or could it be breached during the process? I think this is an ideal study where the authors could potentially discuss their results in the light of the recent arguments on what is causing dissolution – e.g. Peck et al. (2016) stating that this only happens under the breach of the periostracum, while other authors claimed that dissolution is not indiscriminately linked to breached periostracum, occurring sporadically throughout the shell. I would encourage the authors to search for the evidence that can potentially bring more insights into divided views and present an extended discussion on this.

The results in Figure 7 are not consistently analyzed to include all possible comparisons within and between the tows. What is missing is the variability per each tow, and comparison between ALL the tows (not just in the selected few as done now). Once all the additional analyses have been performed, the authors need to discuss the inter- and intra-sample variability in dissolution in much more details – there is obvious a lot of variability on the individual level – why is that? Also, there is not much variability on the carbonate chemistry level, yet quite a lot of difference in shell dissolution between Nov and Dec sampling – provide an explanation for this. Is this due to cumulative exposure? Is dissolution function of size of the animal? In the light of examining dissolution through the exposure point of view, I want to see dissolution as a function of cumulative exposure, not just of in situ omega.

One poorly integrated aspect of this study are shell measurements. How many individuals were analyzed for each bar in Figure 4? I would like to see correlation between shell dissolution and size – per time and per each tow. I suggest the size of each individual with dissolution to be presented in Figure 7. This is the way to demonstrate if/how the size of the organism matters and it can also potentially show if the damages have occurred in the period before sampling. Discuss the implication of these results! Also, correlated shell size to variability in carbonate chemistry conditions (pH, omega) and most importantly, how does cumulative exposure impact the size? These are all the questions that this study can potentially address but have so far failed to do so.

As calculated, density calculations only refer to the density of the organism in the subsample. I would like to see abundances calculated instead (m-2) as it makes it comparable with other studies. As above with shell dissolution, make density correlation with carbonate chemistry (omega, pH) and shell size.

The discussion is poorly written. Instead of discussing their own results, the authors are presenting the results of the other studies –this belongs to the introduction. The discussion needs to be completely restructured and more cohesive.

In the discussion, the authors are referring to the experimental results . . .Laboratory experiments conducted on this same research project showed that holding Limacina in high pCO2 (900 $\mu$atm) for 14 days resulted in significant shell dissolution. ' Where is this statement coming from and how come the authors do not choose to present the results? The only way that the results from natural environment are confirmed is by experimental study, and for this, the results need to be clearly presented!

Also, explain this statement: Our own studies of juvenile pteropods found that L. h. antarctica exhibited a differential response to pCO2 only under ambient temperatures

(-1.0 °C) What sort of differential responses are considered? Explain and corroborate!

---

## Short Comment (SC1) · 6 Feb 2017

To my knowledge there is one paper in the literature showing the impact of high CO2 on shell fitness in the Ross Sea.

MANNO, C., SANDRINI, S., TOSITTI, L., & ACCORNERO, A. (2007) First stages of degradation of Limacina helicina shells observed above the aragonite chemical lysocline in Terra Nova Bay (Antarctica). Journal Marine System

To mention this paper in the introduction and discussion will give a clear overview of the specific research background in this region and will potentially help in the interpretation

of results.

---

## Referee Comment (RC3) · Anonymous Referee #3 · 11 Feb 2017

General comments: This paper is an important contribution to the information being gathered on pteropods throughout the world, and in particular, in the Southern Ocean. Pteropods are at risk due to ocean acidification and any changes in their abundance or distribution would have substantial flow on effects in ecosystems and economically (fisheries). This paper follows on from the work of Bednarsek in investigating dissolution by imaging the surface of shells using SEM. However, the sample preparation methods in this work are far less invasive than some processes used in other studies. I recommend that this paper is published after revisions, particularly in the discussion, as outlined below.

[Figure]

Specific comments: Can the authors explain why they focussed on juvenile pteropods as this may be interesting and add to paper?

Why did the authors use different concentrations of ethanol, between 50% and 90%? I'm not a chemist, but with 50% ethanol, does that mean the other 50% is H2O and could thus cause shell dissolution? Was this investigated? Regardless, please explain your approach to preparation methods. There has been some work (not yet published) on shell damage due to use of H202 bleaching so it would be good to know if you've investigated this too.

The introduction should include a more detailed description of the shell layers - outer prismatic, thick crossed-lamellar layer and inner prismatic - rather than only referring to these in Fig 6 caption. Informing the reader of these layers early will aid in your discussion of the SEM results.

I consider the number of samples you have managed to obtain and study as sufficient. Due to their small size and fragility it is difficult to obtain enough for a thorough analysis - well done!

The authors don't clearly specify why they have chosen to investigate the first whorl only. What was the dissolution near the growth edge like?

Can you explain more clearly why the specific 6 week period was chosen, and what relevance or link the 6 week chemistry results have with the SEM data. That is, how does the chemistry relate to the shell dissolution, if at all, in that short 6 week period?

Further, the discussion/conclusion section should be re-written to focus on and really draw out information from the results. The authors tend to go straight to discussing others' work, rather than giving their results the emphasis they deserve. Elaborate on your results more, then link to other studies.

Technical comments: Line 15: should be "bleached" rather than "breached" when referring to sample preparation methods. line 19: should be "known" rather than "know".

---

## Author Comment (AC1) · 16 Mar 2017

To the Editor:

Thank you for the opportunity to address the reviews of our BGD manuscript that were received during the comment period. We are most grateful for your efforts on our behalf as I am aware of the time editorial duties extract from ones own time budget. To begin, on the following pages, you will find individual responses to the comments and suggestions from the 3 referees. We have done our best to address each of them, and in most cases, we were able to address the query or comment. There are a couple of

instances where we are not quite sure of what is being asked, but again, we have done our best. Overall, we feel we can address the issues raised. We addressed issues regarding the protocol we used to analyze our collected pteropod shells; some referees noted this method is not consistent with some approaches that have been used in past studies of pteropods and their shells. We have addressed this issue in our comments and would note here that we have always followed the example of Bednaršek and her colleagues, as they are very much the world-experts on this topic. When we have made minor departures from these protocols we have done our best to explain and support our decisions. Three other general issues are worth noting – (1) There was perhaps a degree of general confusion where some referees expected controls to be included in the study, or perhaps interpreted this study as comprising experiments. However, the data we present were gathered from a field campaign where the samples where collected via plankton tows at our field site, a single station on the sea ice in McMurdo Sound. We have tried to clarify this and regret causing any confusion on this point. And (2), we received some criticism about a poorly designed "experiment" and here I interpret this to mean that we should have better sampling protocols etc. One issue for us was that these plankton tows were performed at a remote site on the sea ice and we traveled there via snow machine which involved a lot of exposure to the elements. In practice, we were occasionally prevented on sampling on days when the weather did not permit (i.e., -50 °C wind chill and zero visibility, conditions under which the U.S. Antarctic Program senior staff does not allow scientists to leave the field station as a safety precaution). Again, we regret that we were not able to collect in a more consistent manner, but did the best that conditions in the field allowed. Lastly, we can indeed add the paper from Manno et al. to our paper – this came in as a separate comment during the review period. Thank you for your attention to the review of our manuscript and we look forward to hearing from you regarding ours responses to referees.

Best regards, Professor Gretchen Hofmann

  Responses to Referees 1-3 – Comments from referees are listed first, followed by our responses marked as Authors.

Referee 1

1.) I think the conclusion in the abstract mentioned above is not supported by the data because the authors don't know anything about the recent carbonate history of the pteropods but only assume that pteropods lived at under-saturated conditions and this being the reason for the dissolution patterns found.

Authors: We will clarify that statement in the abstract. And, as with many/most pteropod collections made in the field, it is impossible to know the history of exposure of the animals in situ. Thus, we agree that for these collections we do not have a complete record of the carbonate history that these pteropods have experienced. As background, recent publications have identified aragonite saturation states of 1.1 – 1.3 to be sufficient to induce shell dissolution, but with compensatory calcification occurring (Bednaršek 2017). In addition, within the McMurdo Sound autonomous pH sensors deployed in near-shore waters from 2012-2013 recorded aragonite saturation states as low as 0.96 in June and 1.17 in November (Kapsenberg et al. 2015). Throughout the course of these collections, we have provided co-collected water chemistry with aragonite saturation states as low as 1.159 as a means to frame what we could about the environmental exposure of the animals.

2.) The shell dissolution was analyzed only in the protochonch area of the shells. The protoconch is the embryonal/veliger stage and oldest part of the shell. In many species it is naturally shed or broken off from the permanent adult shell (Lalli and Gilmer 1989). I.e. it is the area where damage due to any kind of reason first becomes apparent. No characterization of dissolution of the rest of the shell was done which makes it rather impossible to compare these data with other published work. In fact, the younger part of the shells (juvenile) were apparently not affected by degradation (Fig. 6) indicating pristine shells, thus rather suggesting life at super-saturated Ωaragonite conditions.

Authors: We chose to look at this area of the shell as it was easy to define and was, we thought a good recorder of life-long exposure in situ, i.e., since the protoconch is the oldest part of the shell, it provides the longest record of exposure to undersaturated conditions. By focusing on this small region of the shell it was our intention to standardize the amount of dissolution present between all collections. From a practical perspective we also had issues standardizing area across the entire region of the shell, and additionally that many SEMs was admittedly cost-prohibitive. Notably, a paper recently came out that also focused in part on this region of the shell; here Peck et al. (2016) reported 85% of individuals with dissolution, contained dissolution in the central whorl. In the end, we feel this method worked for us because we were able to identify individuals with pristine shells in addition to individuals with significant dissolution.

3.) No attempt was made to relate biogeochemistry data to dissolutions patterns found, i.e. shell dissolution results were analyzed isolated from prevailing pH, âḐ̌aragonite or pCO2. That way it is not possible to draw any sound conclusion whether the described dissolution patterns have anything to do with carbonate chemistry and even more so with potential changes due to anthropogenic impact. However, as presently prepared, the manuscript "pretends" that results proved a connection between shell dissolution and in situ seawater chemistry. I don't think it allows for such conclusion.

Authors: It was not our intent to draw too tight of a connection between the carbonate chemistry to the shell dissolution measurement. Rather, we strived to provide both sets of data to aid in the readers understanding of the carbonate system at the time of collection so that it is clear the observed dissolution had occurred at some earlier part of the organisms life history. In the conclusion we addressed this with the statement: "In the present study, more than half the shells we examined displayed shell irregularities, but it is unclear as to what depth or season caused the observed dissolution of juvenile shells." We can of course, work harder to make this clear, but we never intended to claim we knew carbonate chemistry conditions at all times when the pteropods were in situ.

4.) Abundance and shell size data were collected but apart from the fact that they were collected and presented in the results, the reader is left alone with these data. Certainly it is not possible to deduce a whole life cycle from such data collected over a rather short time interval (6 weeks), but still it is possible to draw some conclusions and compare with current knowledge.

Authors: We will clarify why we consider these individuals to be juveniles. These data are some of the first collections of pteropods from McMurdo Sound that have occurred with regular sampling at a fixed location. As Referee 1 has noted 6 weeks of data is not sufficient to deduce the life cycle, especially since there is a proposed life-cycle for L.h. antarctica that was constructed from 9 different oceanographic cruises in the Southern Ocean Bednaršek et al. (2012). From these data we called these animals "juveniles" and used the standard practice that others use to note that shell size is used to estimate age and life stage. Further, while we did score these individuals as juveniles we feel we do not have sufficient data to speculate on life history or the larval ecology of Limacina in McMurdo Sound. In general, for our study, pteropods in each tow were roughly in the same size class and we observed modest but steady growth over the 6-week period. This observation is an important insight into their life cycle; however, further work would be needed to say much more about this population of pteropods.

Referee 2

1.) Firstly, the major drawback of the paper is the lack of the working hypotheses. That leaves the work more a compilation of different facts, without any proper integration. Authors have to reconsider how to bring the results in the framework that addresses NEW questions and form the hypothesis around.

Authors: We fully appreciate the instructional suggestions from this referee. However, in this case, the study did not have a hypothesis-driven framework, but represents the result of a field collection in the austral spring in the Southern Ocean. Thus, it

is more of an observation, not an experiment or modeling project (with Observation, Experiment and Modeling being the three core aspects of oceanographic science.). We can endeavor to better reflect that in the Introduction of the paper.

2.) Secondly, this is a poorly designed and conceived study, with inconsistent methodological and statistical approaches that cannot be verified. The authors have obviously used a bit different approach in preparing the samples that is different from previous approaches. a. What was the reason for doing this? b. Was a method systematically tested and how was this done for the authors to be confident in the interpretation of their results? c. Did the periostracum stay intact or could this method have caused breaching of the periostracum at any stage or are the authors confident that no additional damaged were produced during the process? d. I would urge the authors to prepare and present additional results on pteropods that did not undergo this treatment but rather standardized treatment.

Authors: We regret that the Referee finds our study to be poorly designed: the U.S. Antarctic program supported our science plan as a 2012 NSF award. Having said that, we would like to explain some of the challenges one experiences when collecting these types of samples in the field in the Antarctic. The data presented were collected during a field campaign and the samples are difficult to obtain and represent one of the few collections of this member of the zooplankton in the south Ross Sea, where McMurdo Sound is located. To begin, we were not able to consistently sample at our study site on the sea ice as weather often prevented travel. Thus, the sampling interval looks a bit unusual as it was set by access to the station. Second, for these collections, we endeavored to use the appropriate technology, in this case with a special plankton net that was designed to be deployed through fast ice. The net used has been well tested in Antarctic marine field studies (see Sewell 2005), and has been used in numerous studies of Antarctic zooplankton and meroplankton. In terms of the choices regarding shell preparation and SEM techniques, we consulted with several experts on mollusk shell biomaterials and also with pteropod experts. We ultimately chose an approach

and method described by Bednaršek et al. (2012). We regret that this approach was not deemed appropriate by Referee 2 but we felt we were using a published and vetted process. There was one area in which we modified this protocol – namely, we excluded the 6% $H_2O_2$ rinses and plasma etching steps prior to sputter coating the shells. The use of these methods is meant to remove both abiogenic crystals and the periostracum from the surface of the shell. In conversations with experts in this area, we felt that both processes might further damage the shell, and we wanted to image the shells as soon as we could, and also in as natural a condition as one can obtain for SEM. Notably, these steps were also not used in other studies on pteropods (see Peck et al. (2016)). One other note, Dr. Nina Bednaršek, the leader in this field has published a response to the Peck et al. (2016) paper wherein she has gone to great lengths to defend her methods, and provided evidence that measuring dissolution with an intact periostracum is possible, but labor intensive:

"periostracum-covered individuals photographed with the SEM do not display obvious signs of dissolution unless the shell is meticulously examined under high magnification"- Bednaršek et al. 2016

With our samples, the lack of abiogenic crystals on the shell surface indicates that we were successful in cleaning the shells while leaving the periostracum intact, something we did with multiple washes rather than a chemically harsh treatment. Due to the high magnification required to measure dissolution with an intact periostracum we then focused our efforts to a standardized portion of the shell-the central whorl. Finally, due to the remote Antarctic location of this study, and the limited number of individuals available for us to analyze, we are unable to prepare more individuals so that we could compare the results of the $H_2O_2$ treatment. In the end, we did our best to follow the state-of-the-art treatment of the shells in preparation for SEM.

3.) The authors did not have any controls in this study – I request for all the controls to be consistently presented in the paper.

Authors: We regret that we are not clear on what controls the Referee 2 has requested. If by "controls" they would like are related to shell preparation method then, at this stage, we cannot produce such controls. However as noted above, we endeavored to use best practices developed by several notable experts with papers in high-quality peer-reviewed journals. Specifically, there are currently 3 methods in the literature to date presented by Lischka et al. (2012), Bednaršek et al. (2012), and Peck et al. (2016). We followed the methodology set forth by the single most cited paper that describes shell dissolution - Bednaršek et al. (2012). We chose this course because: (i) Dr. Nina Bednaršek is the leading expert in shell dissolution, (ii) at the time of collection (2014) the methodology set forth by Peck et al. (2016) had not been published, and (ii) because of the remote nature of this work transporting non-fixed organisms from the Antarctic, through international customs and to our laboratory in Santa Barbara, CA.

4.) I am perplexed on why the temperature of the water was not measured along with pH. Using temperature from the multi-year series for calculating carbonate chemistry parameters can have major drawbacks and error propagations. Given that temperature can significantly impact aragonite saturation state, it is an imperative to include standard error and uncertainty range in all carbonate chemistry parameters throughout the whole season of measured pH.

Authors: Due to the extreme weather conditions associated with Cape Royds, temperature probes were not taken into the field. While we agree that it would have been better to have temperature data, we are confident in the estimated temperature values we have presented here. This is in part to the fact that the autonomous pH sensors were deployed and actively recording temperature (Cape Evans, unpublished data) throughout the course of these collections. In addition, the water column in this region of the Ross Sea is highly stable, thermally, and rarely varies from -1.9 °C. This has been measured by us in recent sensor deployments (Kapsenberg et al. 2015), and by others (Hunt et al. 2003). Given that our results using these temperatures and method of parameterizing the carbonate system for other studies (e.g., Kapsenberg et

al. 2015) have stood up to peer-review in quality journals (Limnology & Oceanography, Scientific Reports and Environmental Science & Technology), we hope that the editors and referees would deem our methods here sufficient to present analysis of bottle samples. Here, we are only presenting observations, and, we would humbly submit, are not attempting to report on the precise pH or saturation state of these waters as chemical oceanographers might do.

5.) In regards to dissolution estimates, I cannot trace the tows to the carbonate chemistry conditions. Make that link more explicit. I wonder how the authors explain excessive presence of Type III dissolution when the organisms have been exposure to supersaturated conditions throughout the whole study. Is it possible that the damages actually originated before the samples were collected – during the austral winter? If that was the case then the carbonate chemistry presented does not really matter.

Authors: As noted above, we did not intend that the reported chemistry would reflect their entire experience, merely that the presented carbonate data represented the seawater condition at the time of collection. We have not attempted to link the current carbonate chemistry system to dissolution as this damage most likely occurred previously in the austral winter. Nevertheless, we argue that the carbonate chemistry associated with these collections is of value and should be reported. Very few data on carbonate chemistry under fast ice exist, and these data demonstrate what the pteropods are experiencing at these depths in McMurdo Sound.

6.) Where are the figures from Figure 6 from and when where they collected? Indicate how many individuals per each tow look like 6a-c, d-f, g-j. Is image g-j all from one organism?

Authors: These details are given in the main text of the manuscript, but we omitted these details from the figure caption and will make sure to correct this inadvertent omission. For full disclosure, the images presented in Figure 6 are representative images of shells collected in McMurdo Sound. Specifically, there are 2 individuals

represented here, both collected from tow #2 (November 4th, 2014) at the same site that all other collections were made. These two individuals were chosen to represent one individual with no dissolution (a-c,g) and one individual displaying all three type of dissolution (d-f,h-j). Figure 6(g-j) provides higher magnification images as described in the figure legend for figure 6. Briefly, image g is a higher magnification image of image C (individual with no dissolution). Images h-l are higher magnification images of image f (individual with dissolution), highlighting the 3 types of dissolution discussed in this manuscript.

7.) Please, clarify what additional information is captured in Figure 8 that has not been captured in Figure 7. If no different, remove Figure 8.

Authors: We feel that Fig. 8 is a worthwhile figure and adds the perspective that dissolution is rank-ordered phenomenon. Overall, we feel this is a valuable addition to the analysis. To review, Figure 7 provides a per-shell breakdown of the percentage of each dissolution type for each shell in each collection. These data are then compared between tows using a Kurksal-wallis test and Dunn test of multiple comparisons following the previously published method described in Bednaršek et al. (2012). In contrast, Figure 8 presents the number of shells in each tow with the highest level of dissolution (I, II, or III). We believe that this manner of data presentation assists the reader in understanding the number of shells per tow that exhibit Type I dissolution, but not Type II or Type III. For example, Figure 8 shows that while no shells from Tow 14 experienced Type III dissolution, the majority of shells analyzed did experience Type I and II dissolution.

8.) The authors have decided to analyze only the first whorl – why is that? I disagree with the authors that analyzing such a small surface can be unbiased and representative. While this is the easiest approach it does not give the full insight into dissolution. I would like to see the whole shell surface analyzed instead and presented. Only in such way, the results of this study can be comparable to the other studies.

Authors: For this study, we have focused on the first whorl of the shell for two major reasons. First, due to the fact that these individuals are growing throughout the collection period, we standardized the analysis to the protoconch, an area that is roughly the same size across individuals. In addition, we did not focus on the leading edge in part because the carbonate chemistry at the time of collection was super-saturated with respect to aragonite. We would endeavor and/or attempt to analyze the entire shell, although this would likely require 100's more hours of SEM capturing and greatly exceeds what were attempting to communicate in reporting these data. We never intended to compare our patterns of dissolution to those observed in other oceans – namely because, (1) we have no time series data that can capture the environmental history of the pteropods in situ (and no one else does either, we should note), and (2) we are unable to calibrate the age of these wild-caught individuals. Overall, comparative pteropod shell dissolution as a metric of the advancement of ocean acidification, for example, is a long way off. We regret we are unable to present our data in this light, as requested by Referee 2, but remote, expeditionary research often results in data that are a stepping-stone to what one would ideally like to have, in this case, autonomous sensors recording carbonate parameters in a manner that is coupled with pteropod development in natura.

9.) The authors need to provide more discussion about the periostracum. I do not see any evidence that periostracum would be mechanically or chemically damaged – does this then mean that the dissolution is a results of carbonate chemistry conditions, or could it be breached during the process? I think this is an ideal study where the authors could potentially discuss their results in the light of the recent arguments on what is causing dissolution – e.g. Peck et al. (2016) stating that this only happens under the breach of the periostracum, while other authors claimed that dissolution is not indiscriminately linked to breached periostracum, occurring sporadically throughout the shell. I would encourage the authors to search for the evidence that can potentially bring more insights into divided views and present an extended discussion on this.

Authors: We did not see any evidence that the periostracum must be breached for dissolution to occur. We appreciate Referee 2's perspective here, however, this study is not ideal for further investigating this issue. Rather we would argue that a carefully designed and controlled laboratory experiment is needed, as opposed to additional field collections of individuals with unknown life histories. Overall, we humbly submit that our data set falls short of settling the obvious dispute in the literature regarding this mechanism. And to be clear, we had no intention of taking sides in this; we merely hope to present some observations from a hard-to-access, remote part of the Southern Ocean.

10.) The results in Figure 7 are not consistently analyzed to include all possible comparisons within and between the tows. What is missing is the variability per each tow, and comparison between ALL the tows (not just in the selected few as done now). Once all the additional analyses have been performed, the authors need to discuss the inter and intra-sample variability in dissolution in much more details – there is obvious a lot of variability on the individual level – why is that?

Authors: We can definitely make this suggested adjustment in how data in Figure 7 is presented. Presently we show only those interactions that were significantly different as between tows using a Kurksal-Wallis test and Dunn test of multiple comparisons following the previously published method described in Bednaršek et al. (2012). To amend this, we would propose to include the analysis of comparison between all tows in supplementary material. Finally, we have provided the within-tow variation data in Figure 8.

11.) Also, there is not much variability on the carbonate chemistry level, yet quite a lot of difference in shell dissolution between Nov and Dec sampling – provide an explanation for this. Is this due to cumulative expo sure? Is dissolution function of size of the animal? In the light of examining dissolution through the exposure point of view, I want to see dissolution as a function of cumulative exposure, not just of in situ omega.

Authors: Here, Referee 2 has noted that there are differences in shell dissolution throughout the 6- week collection. Referee 2 has suggested examining cumulative exposure or animal size to explain this variability. However, It is our belief that these data collected at a fixed location over time highlight the variability in exposure to under-saturated water experienced by over-wintering juvenile pteropods in Antarctic waters. Furthermore, additional collections throughout a longer window of time would be needed to determine if cumulative exposure explains these results. At present, the only statistically significant differences between tows occurred between tow 14 and tows 7-8. If cumulative exposure was the driving factor, then differences between tows would have occurred in a step-wise fashion.

12.) One poorly integrated aspect of this study are shell measurements. How many individuals were analyzed for each bar in Figure 4?

Authors: We regret omitting this detail from the manuscript. We will correct this in the text and in the figure caption. For clarity, 30 individuals per tow were analyzed for Figure 4.

13.) I would like to see correlation between shell dissolution and size – per time and per each tow. I suggest the size of each individual with dissolution to be presented in Figure 7. This is the way to demonstrate if/how the size of the organism matters and it can also potentially show if the damages have occurred in the period before sampling.

Authors: We fully appreciate this request and would be happy to include the size of each individual with dissolution in the supplementary materials. We avoided presenting the shell dissolution to shell size comparison because throughout the collection period the seawater was supersaturated with respect to aragonite. If we attempted to correlate size to dissolution we would see that December individuals would appear to have a smaller percentage of dissolution then November individuals. We believe this would be a misleading interpretation of these results.

14.) In the discussion discuss the implication of these results! Also, correlated shell

size to variability in carbonate chemistry conditions (pH, omega) and most importantly, how does cumulative exposure impact the size? These are all the questions that this study can potentially address but have so far failed to do so.

Authors: We cannot over-interpret the results that we have. As stated before, we did not intend that the reported chemistry would reflect their entire experience. Merely that it was the seawater condition at the time of collection. We have not attempted to link the current carbonate chemistry system to dissolution as this damage most likely occurred in the austral winter. In reference to discussing the implications of cumulative exposure we referee to our earlier response: It is our belief that these data collected at a fixed location over time highlight the variability in exposure to under-saturated water experienced by over-wintering juvenile pteropods in Antarctic waters. Furthermore, additional collections throughout a longer window of time would be needed to determine if cumulative exposure explains these results. At present, the only statistically significant differences between tows occurred between tow 14 and tows 7-8. If cumulative exposure was the driving factor, then differences between tows would have occurred in a step-wise fashion.

15.) As calculated, density calculations only refer to the density of the organism in the subsample. I would like to see abundances calculated instead (m-2) as it makes it comparable with other studies. As above with shell dissolution, make density correlation with carbonate chemistry (omega, pH) and shell size.

Authors: For this study, we have done our best to analyze our data using previously published methods. In this case, we calculated densities (individuals per m3) based off the previous work in the Southern Ocean (Hunt and Hosie 2006, Hunt et al. 2008). Referee 2 would like to see these densities data correlated with carbonate chemistry. Unfortunately, this data set does not provide enough collection points to accurately correlate density to carbonate chemistry. This is primarily due to the muted change in carbonate chemistry over time. To accurately address this question longer collections across the seasonal alkalization event (Kapsenberg et al. 2015) will be needed. Referee 2 would also like to see these densities data correlated with shell size. Attempting to correlate the collected density to shell size does not provide any significant results. This is driven primarily from two pieces of data: (i) there are 3 size classes and (ii) the three largest peaks in densities were spaced throughout the collection period with each one corresponding to a separate size class.

16.) The discussion is poorly written. Instead of discussing their own results, the authors are presenting the results of the other studies –this belongs to the introduction. The discussion needs to be completely restructured and more cohesive.

Authors: Thank you for this constructive criticism. The goal of this manuscript is to provide a resource for the broader scientific community regarding the distribution and shell condition of Limacina helicina antarctica in McMurdo Sound. We will attempt to re-write the Discussion, but would note that referencing other studies in the Discussion is common practice and was done to help frame our results.

17.) In the discussion, the authors are referring to the experimental results 'Laboratory experiments conducted on this same research project showed that holding Limacina in high pCO2 (900 $\mu$atm) for 14 days resulted in significant shell dissolution. ' Where is this statement coming from and how come the authors do not choose to present the results? The only way that the results from natural environment are confirmed is by experimental study, and for this, the results need to be clearly presented!

Authors: In the Discussion, we mentioned that we have a small set of samples from experiments that were conducted to for analysis of other physiological processes. During these acute exposures to low pH in the lab, we collected a few pteropod shells and imaged them using light microscopy. These images will be used in an upcoming publication, but could be included here to show that this population of Limacina does display dissolution in experiments with acute exposure to high pCO2. We chose not to do this as it would greatly complicate this manuscript and, since the pattern on these shells is impossible to quantify, we would not be able to present quantitative analysis.

18.) Also, explain this statement: Our own studies of juvenile pteropods found that L. h. antarctica exhibited a differential response to pCO2 only under ambient temperatures (-1.0 ◦C) What sort of differential responses are considered? Explain and corroborate!

Authors: We can elaborate on this statement and would do so in a revised version. Notably, for these comparisons of pteropods under different laboratory conditions, we utilized RNA-sequencing to assess differential gene expression, for example.

Referee 3

1. Can the authors explain why they focused on juvenile pteropods as this may be interesting and add to paper?

Authors: The use of juvenile pteropods was a serendipitous occurrence in that we collected at time when juveniles were prevalent. We assume after another 2-3 months, adults would be the dominant form, but by then we would no longer be able to collect them.

2. Why did the authors use different concentrations of ethanol, between 50% and 90%? I'm not a chemist, but with 50% ethanol, does that mean the other 50% is H2O and could thus cause shell dissolution? Was this investigated? Regardless, please explain your approach to preparation methods. There has been some work (not yet published) on shell damage due to use of H202 bleaching so it would be good to know if you've investigated this too.

Authors: The method we utilized here was previously described by Bednaršek et al. (2012) where those investigators suggest fixing samples in 50% ethanol and subsequently transferring the sample to 70% ethanol to avoid the precipitation of abiogenic crystals.

"An alternative approach to minimize or avoid the precipitation of abiogenic crystals on the shell surface during sample fixing would have been to place samples in a low

ethanol grade (50%), followed by subsequent, transfer to higher concentrations (up to 70%) in gradual steps. This method allows dilution of the salts in the high concentrations of seawater and their subsequent removal. Samples should not be fixed in buffered formalin for dissolution studies, because formalin dissolves aragonite crystals." – Bednaršek et al. 2012

Authors: While H2O can be corrosive the fact that 37% of all shells analyzed exhibited no signs of dissolution suggests this short-term incubation in 50% ethanol did not have a dramatic impact on shell condition. In short, we did our best to follow the recommendations set forth in the literature. In addition, we choose not to include H202 bleaching as the ethanol washes were sufficient for the removal of abiogenic crystals from the shell surface.

3. The introduction should include a more detailed description of the shell layers - outer prismatic, thick crossed-lamellar layer and inner prismatic - rather than only referring to these in Figure 6 caption. Informing the reader of these layers early will aid in your discussion of the SEM results.

Authors: We will be happy to expand the description of the shell layers in the introduction of the manuscript.

4. I consider the number of samples you have managed to obtain and study as sufficient. Due to their small size and fragility it is difficult to obtain enough for a thorough analysis - well done!

Authors: Our thanks to Referee 3 for appreciating the challenges of operating in remote field locations under non-ideal conditions. We were happy to get them home as well!

5. The authors don't clearly specify why they have chosen to investigate the first whorl only. What was the dissolution near the growth edge like?

Authors: We would be happy to include a rational for focusing on the first whorl only in a revised version of the manuscript. As mentioned in a Response to Referee 1, we

chose to limit our investigation to the first whorl of the shell as it was easy to define and was, we thought a good recorder of life-ling exposure in situ, i.e., since the protoconch is the oldest part of the shell, it provides the longest record of exposure to undersaturated conditions. By focusing on this small region of the shell it was our intention to standardize the amount of dissolution present between all collections. From a practical perspective we also had issues standardizing area across the entire region of the shell, and additionally that many SEMs was admittedly cost-prohibitive.

6a. Can you explain more clearly why the specific 6 week period was chosen, and what relevance or link the 6 week chemistry results have with the SEM data. That is, how does the chemistry relate to the shell dissolution, if at all, in that short 6-week period?

Authors: We would be happy to expand our explanation in the methods as to why this 6 week period was chosen. We have focused on this specific 6-week period as it was the only window in which we could safely reach the research site approximately 30 miles away from station across seasonal sea ice. McMurdo Station is operated by the U.S. NSF Antarctic Program and NSF limits sea ice access to a small window every year (mid-October to early-December). During this time frame, the ice is thick and stable enough to support travel on the sea ice. Indeed, during the 2014 field season, we were the last group allowed to pass beyond Cape Evans before the ice became unsafe for science travel. Lastly, as mentioned earlier, it was not our intent to draw too tight of a connection between the carbonate chemistry to the shell dissolution measurement. Rather, we strived to provide both sets of data to aid in the readers understanding of the carbonate system at the time of collection so that it is clear the observed dissolution had occurred at some earlier part of the organisms life history. In the conclusion we addressed this with the statement: "In the present study, more than half the shells we examined displayed shell irregularities, but it is unclear as to what depth or season caused the observed dissolution of juvenile shells." We can of course, work harder to make this clear, but we never intended to claim we knew carbonate chemistry conditions at all times when the pteropods were in situ.

6. Further, the discussion/conclusion section should be re-written to focus on and really draw out information from the results. The authors tend to go straight to discussing others' work, rather than giving their results the emphasis they deserve. Elaborate on your results more, then link to other studies.

Authors: We thank the review for this collegial and useful suggestion and will endeavor to improve the discussion of our results in a revised manuscript.

7. Technical comments: Line 15: should be "bleached" rather than "breached" when referring to sample preparation methods. line 19: should be "known" rather than "know".

Authors: We will definitely make these corrections in a revised manuscript.

References Cited

Bednaršek, N., Tarling, G. A., Bakker, D. C., Fielding, S., Cohen, A., Kuzirian, A., McCorkle, D., Lézé, B., and Montagna, R.: Description and quantification of pteropod shell dissolution: a sensitive bioindicator of ocean acidification, Global Change Biology, 18, 2378-2388, doi:10.1111/j.1365-2486.2012.02668.x, 2012.

Bednaršek, N., and Ohman, M. D.: Changes in pteropod distributions and shell dissolution across a frontal system in the California Current System, Marine Ecology Progress Series, 523, 93-103, doi:10.3354/meps11199, 2015.

Hunt, B.P.V., Hoefling, K., & Cheng, C.: Annual warming episodes in seawater temperatures in McMurdo Sound in relationship to endogenous ice in notothenioid fish, Antarctic Science, 15, 333-338, doi:10.1017/S0954102003001342, 2003.

Hunt B.P.V., Hosie G.W.:The seasonal succession of zooplankton in the Southern Ocean south of Australia, part I: the seasonal ice zone. Deep Sea Research I 53:1182-1202, doi:10.1016/j.dsr.2006.05.001, 2006.

Hunt, B. P. V., Pakhomov, E. A., Hosie, G. W., Siegel, V., Ward, P., and Bernard, K.: Pteropods in Southern Ocean ecosystems, Progress in Oceanography, 78, 193-221,

doi:10.1016/j.pocean.2008.06.001, 2008.

Kapsenberg, L., Kelley, A. L., Shaw, E. C., Martz, T. R., and Hofmann, G. E.: Near-shore Antarctic pH variability has implications for the design of ocean acidification experiments, Science Reports, 5, 9638, doi:10.1038/srep09638, 2015.

Lischka, S., Büdenbender, J., Boxhammer, T., and Leibniz-Riebesel, U.: Impact of ocean acidification and elevated temperatures on early juveniles of the polar shelled pteropod Limacina helicina: mortality, shell degradation, and shell growth, Biogeosciences, 8, 919-932, doi:10.5194/bg-8-919-2011, 2011.

Peck, V. L., Tarling, G. A., Manno, C., Harper, E., and Tynan, E.: Outer organic layer and internal repair mechanism protects pteropod Limacina helicina from ocean acidification, Deep Sea Research II, 41-52, doi:10.1016/j.dsr2.2015.12.005, 2016.

Sewell, M. A.: Examination of the meroplankton community in the south-western Ross Sea, Antarctica, using a collapsible plankton net. Polar Biology 28: 119-131, doi:10.1007/s00300-004-0670-9, 2005.